# Radiographical Diagnostic Evaluation of Mandibular Cortical Index Classification and Mandibular Cortical Width in Female Patients Prescribed Antiosteoporosis Medication: A Retrospective Cohort Study

**DOI:** 10.3390/diagnostics14101009

**Published:** 2024-05-13

**Authors:** Keisuke Seki, Maki Nagasaki, Tona Yoshino, Mayuko Yano, Aki Kawamoto, Osamu Shimizu

**Affiliations:** 1Department of Comprehensive Dentistry and Clinical Education, Nihon University School of Dentistry, 1-8-13, Kanda-Surugadai, Chiyoda-ku, Tokyo 101-8310, Japan; 2Division of Dental Education, Dental Research Center, Nihon University School of Dentistry, 1-8-13, Kanda-Surugadai, Chiyoda-ku, Tokyo 101-8310, Japan; 3Department of Oral and Maxillofacial Surgery I, Nihon University School of Dentistry, 1-8-13, Kanda-Surugadai, Chiyoda-ku, Tokyo 101-8310, Japan; nagasaki.maki@nihon-u.ac.jp (M.N.); shimizu.osamu@nihon-u.ac.jp (O.S.); 4Department of Oral and Maxillofacial Surgery, Yokohama City University Graduate School of Medicine, 3-9 Fukuura, Kanazawa-ku, Yokohama 236-0004, Japan; yoshino.ton.hg@yokohama-cu.ac.jp; 5Nihon University School of Dentistry Dental Hospital, 1-8-13, Kanda-Surugadai, Chiyoda-ku, Tokyo 101-8310, Japan; mayu81173@gmail.com; 6Dental Hygienist Section, Nihon University School of Dentistry Dental Hospital, 1-8-13, Kanda-Surugadai, Chiyoda-ku, Tokyo 101-8310, Japan; kawamoto.aki@nihon-u.ac.jp

**Keywords:** bisphosphonate, denosumab, digital panoramic radiography, mandibular cortical index, mandibular cortical width

## Abstract

Osteoporosis is often detected late and becomes severe because of a lack of subjective symptoms. Digital panoramic radiography (DPR) has been reported to be useful for osteoporosis screening based on the morphological classification of the mandibular inferior cortex. The purpose of this study was to evaluate the sensitivity and specificity of the mandibular cortical index (MCI) in the diagnosis of osteoporosis in a group of patients who were and were not using antiosteoporosis medication (AOM). Three hundred and fifty female patients aged 40 years or older who had DPR imaging performed during a 6-year period from December 2015 to February 2022 met the selection criteria. Two examiners recorded mandibular cortical width and MCI from the images. These results were statistically examined together with the patients’ demographic data. Forty-nine patients were using AOM (13 nonbisphosphonate/denosumab and 36 bisphosphonate/denosumab). MCI type 3 was the most common in the AOM group. In the MCI classification, DPR imaging among the AOM group was more sensitive (0.95) than that of the control group. This method of estimating osteoporosis based on MCI classification using DPR images has high sensitivity, especially in patients using AOM, suggesting that this method is useful as a screening test.

## 1. Introduction

Osteoporosis and associated fractures lead to a decline in overall function and loss of independence, reducing patients’ quality of life [1,2,3]. Osteoporosis is a highly prevalent chronic bone metabolism disorder [4,5], affecting an estimated 200 million people worldwide [6]. However, many patients are asymptomatic, which delays diagnosis and prevents them from receiving treatment [7]. In recent years, a dental approach has been attracting attention as a solution to the problem of undiagnosed patients. In the radiological diagnosis of dental diseases (dental caries and periodontal disease), digital panoramic radiography (DPR) is routinely used to obtain a comprehensive view of the jawbone. It has been reported that approximately 90% of subjects with coarse cortical bone morphology and an abnormally shaped mandibular cortex had osteoporosis [8,9]. Because the shape of the mandibular inferior cortex reflects the bone density of the lumbar vertebrae and the femur and is associated with bone metabolism markers and fracture risk [10,11], the evaluation of the mandibular cortical index (MCI) by DPR imaging is a useful screening tool for detecting osteoporosis [12,13,14].

Various bone resorption inhibitors are prescribed to patients to treat osteoporosis and the bone lesions of malignant tumors, and the increased risk of developing medication-related osteonecrosis of the jaw (MRONJ), an intractable hard and soft tissue disease, is an important consideration in the dental field [15,16,17]. The dental treatment of patients who use such medications is often complicated. To prevent MRONJ when undertaking dental treatment, it is important to know whether the patient is receiving antiresorptive therapy such as bisphosphonates (BPs) [18,19] or denosumab (Dmab) [20], an anti-RANKL antibody.

The relationship between DPR imaging and osteoporosis is based on a demographic and anatomical perspective, and few investigations have focused on the dental profile characteristics of patients using antiosteoporosis medication (AOM) [21,22]. In particular, the difficulty and cost involved in ascertaining detailed medication status, such as the type of AOM, duration of use, and periods of drug holiday, have been pitfalls in this research field. Thus, there are many unknowns regarding patients using high-risk AOM who should be given special attention in dental treatment, and there is a great lack of evidence for predictive dental treatment.

Many bone resorption inhibitors act on bone metabolic turnover and have a significant medical benefit in preventing bone fractures. However, there is little information on how the use of these drugs affects the mandible and how the radiological findings of the morphology of the mandibular inferior cortex are altered. Patients using AOM may exhibit radiological changes such as increased calcification; therefore, the clinical question remains as to whether screening for osteoporosis using DPR images may result in a false-negative diagnosis. To address this gap in the evidence, epidemiological studies based on detailed individual observation are essential.

Familiarity with the mandibular morphological and imaging characteristics of the increasing number of patients using AOM will be useful for medical professionals. Dentists can detect changes and signs in radiological images of patients using AOM to prompt a visit to a doctor. This may decrease the severity of lesions, reduce medical costs, and improve the health of patients.

The purpose of this study was to elucidate diagnostic characteristics and correlations using DPR images to test the null hypothesis that “the association between mandibular inferior cortical shape and osteoporosis does not differ between patients using AOM and control patients”. Furthermore, we aimed to detect characteristic radiological findings in patients using AOM to enable better diagnosis. Therefore, an observational study was planned with AOM as the exposure factor and severe MCI classification as the outcome, and a single-center retrospective cohort study was conducted using descriptive epidemiology.

## 2. Materials and Methods

### 2.1. Study Population

A retrospective cohort study was conducted at a dental facility in Mishima, Shizuoka, Japan. This observational study was approved by the Ethics Committee of Nihon University School of Dentistry (Permit No. EP20D006) and was conducted in accordance with the guidelines for observational/descriptive studies on the enhanced reporting of observational studies in epidemiology, in accordance with the 1975 Declaration of Helsinki, revised in 2013 [23]. Patients who visited the Nihon University School of Dentistry Mishima Dental Center between 2015 and 2022 were included in the study, and patient information records were collected as in our previous study [24]. Documents describing the collection, purpose of use, and research methods were posted in the clinic, and the information was disclosed and made known to the subjects through its website.

### 2.2. Patient Selection and Data Sources

Inclusion criteria for this study were (1) patients who visited the clinic between December 2015 and February 2022, (2) women aged 40 years or older at the time of visit, and (3) patients who had digital panoramic radiographs taken for dental treatment. The exclusion criteria were (1) patients whose radiographs did not show the mandibular inferior cortical morphology because of ghosting in the image or poor positioning; (2) patients with a history of bone-destroying events such as mandibular body osteotomy, mandibular reconstruction, or neoplastic lesions; and (3) patients who had received radiation therapy in the head and neck region. The experimental method is shown in Figure 1.

### 2.3. Variables and Data Collection

When extracting the demographic and medical data, examiners were blinded to whether or not the patients were using AOM. Age, body mass index (BMI), smoking, medical history (osteoporosis, malignant tumor, rheumatoid arthritis, hypertension, and diabetes mellitus), use of AOM and duration of use, and history of MRONJ were extracted and recorded from the initial examination records. Patients who used AOM were defined as the AOM group, and patients who did not use AOM were defined as the control group. The AOM group was further divided into two groups: those using BPs or Dmab, which are currently considered a cause of MRONJ (BP/Dmab group) [16], and those using only vitamin D3 or selective estrogen receptor modulators (non-BP/Dmab group). Occlusal forces, which reflect the number of teeth and occlusal condition, have a biomechanical effect on remodeling and maintaining the structure of the mandibular body [25]. For this reason, we recorded the number of teeth present and evaluated the occlusal condition using a simplified version of the Eichner classification [26]. Patients with A1–B1 with three or more occlusal support areas were classified as class 1; those with B2–C1 with two or fewer areas but with occlusal support in the anterior teeth were classified as class 2; and those with C2 without occlusal support areas were classified as class 3. All X-ray images were taken with a ProMax 3D (Planmeca, Helsinki, Finland), an X-ray machine used in daily practice. The imaging conditions were as follows: tube voltage, 66 kV; tube current, 9 mA; and irradiation time, 19 s. The diagnostic imaging software used was the Romexis 2D and 3D module (Planmeca). The output image data were evaluated on a diagnostic monitor (Eizo, Ishikawa, Japan), which allowed the image on the monitor to be scaled up or down and the density and contrast to be changed.

### 2.4. Measurement of Mandibular Cortical Width (MCW)

The cortical bone thickness (α–β) of the mandibular inferior cortex just below the mental foramen was measured by one examiner (K.S.) in increments of 0.1 mm three times on each side using the image analysis software described above (Figure 2A). The average of all measurements on both sides was taken as the representative value of the MCW.

### 2.5. Measurement of MCI

The morphology of the mandibular inferior cortex was classified into three types, as described by Taguchi et al. [9] (type 1, smooth inner surface of cortical bone; type 2, irregular inner surface of cortical bone with linear resorption; type 3, severe linear resorption and cortical bone rupture over the entire cortical bone) (Figure 2B–D). Evaluation was performed bilaterally, and the worst assessment was used. Diffuse opacities such as sclerosing osteomyelitis that did not correspond to any of the above types were recorded separately. MCI was assessed twice each by two dentists (K.S., a periodontist, and M.Y., a dental trainee) who were trained in classification beforehand, and the second assessment was used. The results of the two examiners were then compared; in the case of the same rating, the same rating was adopted, and in the case of a difference, the more severe rating was used as the final MCI value.

### 2.6. Interrater Reliability

The two examiners performed MCI classification on 40 randomly selected panoramic X-rays and classified them again in a different order 1 week later. Cohen κ scores were obtained for intra- and interindividual reproducibility. Kappa scores between 0.41 and 0.60 indicated fair agreement, between 0.61 and 0.80 indicated good agreement, and between 0.81 and 0.92 indicated very good agreement [27].

### 2.7. Statistical Analysis

Statistical analyses were performed using EZR (Saitama Medical Center, Jichi Medical University, Saitama, Japan), a graphical user interface of R (version 4.0.0, The R Foundation for Statistical Computing, Vienna, Austria) [28]. The sample size calculation was based on the number of patients diagnosed with osteoporosis at hospitals, with an α error of 0.05, a power of 0.8, and a sample ratio of 1:0.12 for the two groups. Using the demographic data of the AOM and control groups obtained by descriptive statistics, the statistical differences for each variable were examined. The Kolmogorov–Smirnov test was used for normality of data distribution for continuous variable outcomes (age, BMI, number of teeth present, and MCW), and *p* ≥ 0.05 was considered a normal distribution. The F-test was conducted for each continuous variable. Welch’s *t*-test was conducted only for MCW values (*p* = 0.03) where equal variances were not found in the data of the two groups, and Mann–Whitney’s *U* test was conducted for other continuous variables that did not follow a normal distribution. For categorical variables (concomitant systemic diseases), the chi-square test was employed if the overall number of cases was ≥40, and Fisher’s exact test was used if the overall number of cases was between 20 and 40 and the expected value was <5. Furthermore, for MCW, an analysis of variance test was performed for the two AOM subgroups (BP/Dmab group and non-BP/Dmab group) and the control, for a total of three groups, and a Tukey post hoc test was performed.

The more severe type 2 and 3 cases were dichotomized as the severe category and type 1 as the normal category; then, the Cochran–Armitage trend test was conducted to examine the correlation between MCI and the two categorical variables (AOM use; occlusal status), and a contingency table was created. All tests were statistically significant at *p* < 0.05.

## 3. Results

### 3.1. Patient Demographics and Medical Data

The patients’ data are shown in Table 1. Of the 587 female patients aged over 40 years who visited the hospital during the study period, 352 had received DPR imaging. Of these, one patient had undergone a segmental mandibulectomy, and one patient’s mandible could not be evaluated because of poor positioning, resulting in a total of 350 images for final evaluation. Of the 49 patients using AOM, 36 were in the BP/Dmab group, and 13 were in the non-BP/Dmab group. Eight MRONJ cases were observed. Taking those diagnosed at the medical hospital as the reference standard, there were 46 patients with osteoporosis, with a prevalence of 13.1%. The control group consisted of 301 patients who were not using AOM.

The mean age of the AOM group was 76.0 years (48–94 years; median, 78 years), which was significantly older (by 21.1 years) than the control group. The mean BMI was 20.4 (14.1–26.2; median, 20.2) in the AOM group and 21.3 (15.1–37.6; median, 20.7) in the control group, with both groups falling within the normal weight category by obesity criteria. Smoking was more prevalent in the control group (10.3%), but the difference was not significant. The mean number of teeth was significantly higher in the control group (mean, 25.5 teeth) than in the AOM group by approximately five teeth. The simplified Eichner classification had the highest number of class 1 (55.1%) and the lowest number of class 3 (12.2%) cases. Among the AOM group, osteoporosis (83.7%), hypertension (51.0%), and malignancy (22.4%) were the most common comorbidities, followed by rheumatoid arthritis (10.2%), which was the least common. All of these comorbidities were significantly more common than in the control group (all less than 6%). Five untreated osteoporosis patients were observed in the control group. There was no significant difference in diabetes between the two groups.

In the AOM group, 73.5% used BP/Dmab, and 26.5% used non-BP/Dmab drugs. Oral BPs (mean duration of medication, 41.3 months) were the most commonly used AOM (Table 2). Eight patients in the AOM group had a history of MRONJ.

### 3.2. MCW Comparison

The mean MCW value was significantly larger in the control group (2.86 mm) than in the AOM group (2.54 mm) (*p* = 0.0096). Multiple comparisons between the two subgroups and the control group showed a significant difference between the non-BP/Dmab group (2.39 mm) and the control group (2.87 mm).

### 3.3. Evaluation of MCI

The Cohen κ score for intraindividual reproducibility was 0.73 for examiner 1 (K.S.), 0.64 for examiner 2 (M.Y.), and 0.60 for interindividual reproducibility. Within the AOM group, 6.1% were classified as type 1 MCI, 38.8% as type 2, and 55.1% as type 3 (the most common type). Within the control group, 16.3% were classified as type 1, 22.9% as type 3, and 60.8% as type 2 (the most common). Eight patients exhibited sclerosing osteomyelitis-like findings that could not be evaluated, but all of these patients were able to be classified on the opposite side. There were significant differences between the two groups for all types of MCI. The correlation between the severity of MCI and the AOM subgroups was examined by the Cochran–Armitage trend test (Table 3), and the type or treatment of AOM did not correlate with the MCI category (*p* = 0.07). However, the simplified Eichner classification was correlated with the MCI category (*p* < 0.01). A contingency table for diagnosing osteoporosis from MCI types 2–3 is shown in Table 4. The sensitivity of the AOM group was high (0.95), and specificity was low (0.13). However, compared with the overall results, the positive predictive value was higher (0.85), and the negative predictive value (0.33) was lower.

## 4. Discussion

This was a single-center, retrospective cohort study focusing on dental radiographs of patients using AOM and examining the relationship between mandibular inferior cortical morphology and osteoporosis. The design of this study assumed osteoporosis as the main disease. Because of its high prevalence in women [1,29], women in their 40s, who were considered to be at the beginning of menopause, were included in the study. The study cohort comprised patients who presented to the same facility. This was meant to reduce intergroup bias, such as that associated with the surgeon or examination procedure. The results showed a large difference in mean age between the AOM and control groups. Improvements in this cohort design will be needed in future studies because similar ages when comparing groups of patients will lead to more accurate experimental results. Patients in both groups recorded BMIs within the healthy range, indicating an average body size. The large number of smokers in the control group reflects the high smoking rate among young people [30]. In the AOM group, BPs or Dmab were used for three-quarters of the treatments, with oral BPs being the most common. This was a similar finding to recent MRONJ survey results in Japan [31]. Oral drug use in older adults is problematic because of their poor medication compliance [32]. In this study, compliance was not investigated, and the exact medication status of the patients could not be ascertained. The fact that there were 8 MRONJ cases out of 49 patients treated with AOM differs significantly from previous studies [16,17,18,19,20] that reported low incidence rates. However, this is presumably because patients were referred from local practices to our more advanced dental center [33,34]. These results should be interpreted with this background in mind.

Previous experimental studies in rats treated systemically with alendronate (a BP) have reported increased bone mineral density and increased maximum fracture force [35]. However, the long-term use of alendronate has been shown to increase bone mineralization, decrease collagen content, and reduce the capacity for bone remodeling [36], so the conclusions are still divergent. Therefore, we focused on MCW and MCI in this study to observe the effect of AOM treatment on the human mandible.

Although previous studies on MCW have revealed racial differences [37], in Japanese patients, MCW is reported to change with age, with a decrease in bone mineral density and an increased risk of osteoporosis <2.6–3 mm [14]. In the present study, the MCW in all groups was <3.0 mm and was significantly lower in the non-BP/Dmab group than in the control group. This implies that AOMs other than BPs and Dmab do not affect the change in MCI, which decreases with age. However, the difference between the experimental groups in this study was only at the 0.1 mm level. Although a computer-aided diagnosis system can accurately measure MCI, it is difficult to apply this small difference to clinical diagnosis because it is assumed that, in actual clinical practice, MCI is often measured by visual inspection.

When comparing the MCI between the two groups, the differences between the types were first examined, and significant differences were found between all types: types 1 and 2 were more common in the control group, while type 3 was more common in the AOM group. Types 2 and 3 are considered to present a high risk of osteoporosis, and the correlation between AOM treatment and occlusal status was examined by the Cochran–Armitage test in all the subjects. This test method is suitable for examining correlations of categorical variables rather than continuous variables. The results showed a correlation between the Eichner classification severity and MCI values. This suggests that there is a correlation between the worsened occlusal condition and the severity of the MCI. The mandibular cortical shape was also worse when accompanied by poor occlusion, a finding similar to that reported in a previous study [38]. This finding suggests that the morphology of the mandibular inferior cortex is expected to be obscured in DPR images of older edentulous patients and that we need to pay special attention to the prevalence of osteoporosis.

Finally, the diagnostic characteristics of MCI types 2–3 and osteoporosis were evaluated for the AOM group. The sensitivity was 0.95, indicating that this test could rule out osteoporosis in type 1 cases. The specificity was 0.13; therefore, the osteoporosis was less likely to be overlooked but also more likely to be wrongly diagnosed as osteoporosis. When the cutoff was changed and only MCI type 3 (*n* = 27) was examined, the sensitivity and specificity changed significantly (0.56 and 0.50, respectively). Thus, the sensitivity and specificity depend on the cutoff value. Although sensitivity and specificity are inversely related, and it would be ideal if both were close to 100%, the present results suggest a tendency toward overdiagnosis. The higher positive predictive value (AOM group = 0.85) compared with all subjects (total = 0.15) was related to the greater prevalence of osteoporosis (0.84) within the AOM group. Unlike the predictive value, because the sensitivity and specificity are not affected by the prevalence rate, they are considered to represent the characteristics of the test itself. These results suggest that a screening test for osteoporosis using DPR imaging, which is less invasive and less costly than other diagnostic methods, is useful for exclusion diagnoses even in patients using AOM, although it is necessary to confirm the diagnosis with a highly specific test. Because AOM significantly improves fracture risk, it was predicted that the mandibular morphology of patients using AOM would appear to be healthier. Therefore, one of the objectives of the present study was to confirm any changes. In the present study, however, there was no evidence for the increased calcification of the mandibular cortex, even in patients using AOM. Although not investigated in this study, bone-modifying agents are sometimes given to patients receiving steroid therapy, but it is possible that a decrease in bone mineral density is not observed, which should be clarified in future studies.

Taken together, the results obtained in this study reject the null hypothesis that “the association between mandibular inferior cortical shape and osteoporosis does not differ between patients using AOM and control patients”. Our findings suggest that the diagnostic screening method for osteoporosis using the MCI classification is particularly sensitive in patients using AOM. Osteoporosis tends to be detected late as a “silent disease,” but the increase in the number of patients with undiagnosed osteoporosis who are screened by DPR imaging, which is frequently used in dental treatment, and who are referred to specialized medical institutions will be of great benefit in improving their health. From this point of view, although the patients using AOM in this study are considered to have already started osteoporosis treatment and are under health management, the dentist may be able to detect signs of clinical changes, such as increased severity of osteoporosis symptoms and changes in disease status. Sharing the results of this study among medical professionals will assist in providing accurate patient information to physicians, orthopedic surgeons, and gynecologists when invasive dental procedures such as tooth extraction and dental implants are planned for older patients.

A strength of this study is that it highlights the validity of the MCI classification, which is considered useful as a screening test for osteoporosis, for the increasing number of people using AOM in the future. Although screening diagnostic methods using the MCI were discovered approximately 20 years ago, few studies have examined imaging findings in AOM users. This is a novel aspect of our study, and the results provide missing information in the field of osteoporosis treatment. However, one limitation of this study is that it was a single-center study, and the general application of the results should be interpreted with caution. In other words, the single-center design may not provide a representative sample of the general population, thus limiting the generalizability of the findings. Future studies should solve this problem with a longer experimental duration. To obtain more comprehensive clinical results, a multicenter survey is needed and will be the subject of future research. Additionally, to examine the pure effect of osteoporosis alone, it will be necessary to exclude patients taking AOM for non-osteoporotic purposes and patients with other metabolic diseases.

In conclusion, the MCI classification of DPR images of the mandibular inferior cortex in female patients over 40 years of age taking AOM revealed that type 3 was the most common. This diagnostic method requires careful determinations of cutoff values using the severity of the classifications. New findings show that changes in the cutoff value affect the sensitivity of the test. Our findings suggest that dentists could provide more reliable information about osteoporosis to the medical community in the health management of patients using AOM.

## Figures and Tables

**Figure 1 diagnostics-14-01009-f001:**
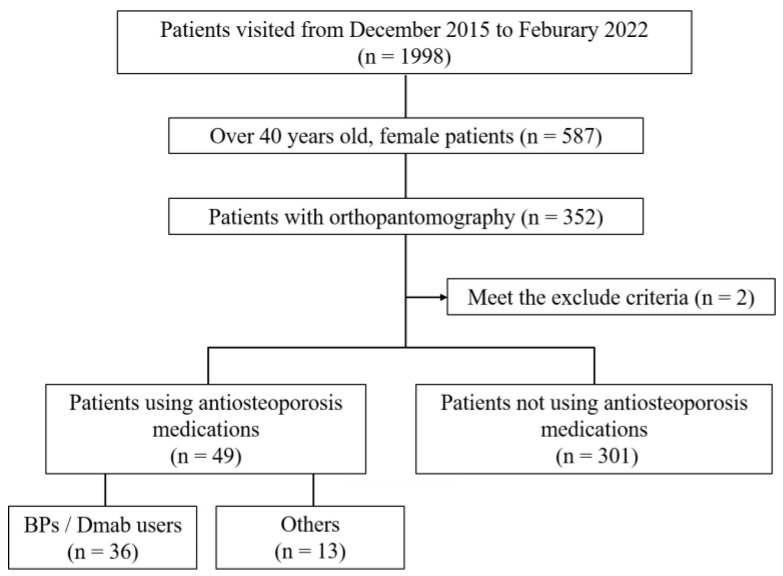
The experimental procedure in patients using antiosteoporosis medications. BP, bisphosphonate; Dmab, denosumab.

**Figure 2 diagnostics-14-01009-f002:**
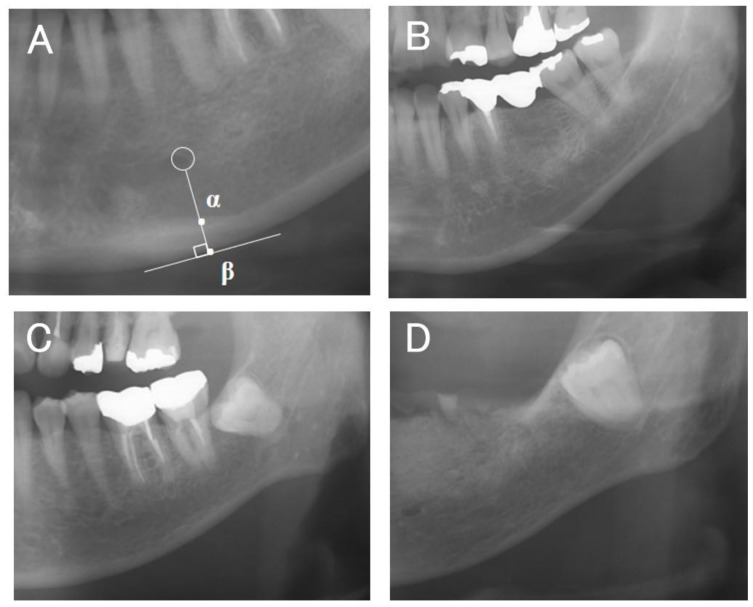
(**A**) The mandibular cortical width (MCW) was determined by drawing a perpendicular line from the mental foramen (circle) to the inferior margin of the mandible and measuring the distance between α and β. (**B**) Typical example of type 1 (normal category). (**C**) Typical example of type 2. (**D**) Typical example of type 3 (types 2 and 3 are the severe category).

**Table 1 diagnostics-14-01009-t001:** Demographic and patient characteristics for the antiosteoporosis medication (AOM) group and the control group.

	AOM Group (*n* = 49)	Control Group (*n* = 301)	Significant Difference
Age (years) ^a^	76.0 ± 10.1 (median 78)	54.9 ± 12.1 (median 53)	**
BMI (kg/m^2^) ^a^	20.4 ± 2.4 (median 20.1)	21.3 ± 3.3 (median 20.7)	ns
Smoking ^b^	2 (4.1)	31 (10.3)	ns
Present teeth ^a^	20.4 ± 7.0 (median 22)	25.5 ± 5.7 (median 27)	**
Comorbidities * overlapping			
Osteoporosis ^c^	41 (83.7)	5 (1.7)	**
All types of malignancies ^b^	11 (22.4)	15 (5.0)	**
Rheumatoid arthritis ^b^	5 (10.2)	5 (1.7)	**
Hypertension ^b^	25 (51.0)	0 (0)	**
Diabetes ^b^	7 (14.3)	17 (5.6)	ns
Antiresorptive therapy			
BP/Dmab	36 (73.5)	0	
Non-BP/Dmab	13 (26.5)	0	
MRONJ	8	0	
Eichner’s classification (simplified type) ^c^			
A1–B1: Class 1	27 (55.1)	266 (88.4)	**
B2–C1: Class 2	16 (32.6)	25 (8.3)	ns
C2–C3: Class 3	6 (17.2)	10 (3.3)	**
MCW (mm) ^d^	2.54 ± 0.80 (median 2.62)	2.86 ± 0.64 (median 2.88)	**
BPs/Dmab (*n* = 36)	2.60 ± 0.81 (median 2.68)		
Non-BP/Dmab (*n* = 13)	2.39 ± 0.75 (median 1.97)		* with control group ^e^
MCI ^c^			
Type 1 (normal category)	3 (6.1%)	49 (16.3%)	**
	BP/Dmab 2, non-BP/Dmab 1		
Type 2 (severe category)	19 (38.8%)	183 (60.8%)	**
	BP/Dmab 16, non-BP/Dmab 3		
Type 3 (severe category)	27 (55.1%)	69 (22.9%)	**
	BP/Dmab 18, non-BP/Dmab 9		

Mean ± S.D., or *n* (%), ^a^ Mann-Whitney *U* test, ^b^ Fisher’s exact test, ^c^ Chi-squared test, ^d^ Welch’s *t* test, ^e^ ANOVA test. AOM, antiosteoporosis medication; ns, not significant; BMI, body mass index; BP, bisphosphonate; MRONJ, medication-related osteonecrosis of the jaw; MCW, mandibular cortical width; MCI, mandibular cortical index; ns, not significant; * *p* < 0.05; ** *p* < 0.01.

**Table 2 diagnostics-14-01009-t002:** Details of antiosteoporosis medications (overlapped).

AOM Group (*n* = 49)		
	Antiosteoporosis Medication	Mean Duration (Months, Minimum–Maximum)
BP/Dmab (*n* = 36)		
	Oral BPs (24)	41.3 (1–212)
	Injected BPs (4)	17.5 (4–29)
	Dmab (7)	32.3 (2–66)
	Sequential therapy (1)	over 12
Non-BP/Dmab (*n* = 13)		
	Alfacalcidol (7)	53.0 (18–96)
	Bazedoxifene acetate (2)	15.5 (13–18)
	Tamoxifen (2)	43.0 (12–74)
	Teriparatide (1)	7
	Trastuzumab (1)	12

AOM, antiosteoporosis medication.

**Table 3 diagnostics-14-01009-t003:** Correlation between mandibular cortical index (MCI) and chronological variables (Cochran–Armitage trend test).

MCI Class 2–3 Case: Severe Category (*n* = 298)		*p* Value	Significant Difference
	Categorical variable		
Antiosteoporosis medications		0.066	ns
Control (*n* = 252)	1		
Non-BP/Dmab (*n* = 12)	2		
BP/Dmab (*n* = 34)	3		
Eichner’s classification (simplified type)		0.008	**
Class 1: A1–B1 (*n* = 242)	1		
Class 2: B2–C1 (*n* = 41)	2		
Class 3: C2–C3 (*n* = 15)	3		

ns, not significant; ** *p* < 0.01.

**Table 4 diagnostics-14-01009-t004:** Contingency table for diagnosing osteoporosis.

	AOM Group/Total (*n*)	
	Osteoporosis	Non-Osteoporosis
MCI Type 2–3 (severe category)	39/44	7/254
MCI Type 1 (normal category)	2/2	1/50
	AOM group (*n* = 49)	Total (*n* = 350)
Prevalence	0.84	0.13
Sensitivity	0.95	0.96
Specificity	0.13	0.16
Positive predictive value	0.85	0.15
Negative predictive value	0.33	0.96

AOM, antiosteoporosis medication.

## Data Availability

The data used in this study are available upon reasonable request by email from Nihon University School of Dentistry, Japan.

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
