# Peer review of "Radiographical Diagnostic Evaluation of Mandibular Cortical Index Classification and Mandibular Cortical Width in Female Patients Prescribed Antiosteoporosis Medication: A Retrospective Cohort Study"

_diagnostics, 2024, doi:10.3390/diagnostics14101009_

Round 1

Reviewer 1 Report

Comments and Suggestions for Authors

The researchers conducted a study to assess the cognitive impairment in female patients who were taking antiresorptive medicine. The study design is not accurately represented by the title. The authors have assessed MCI, MCW and ECI).The title does not indicate this. Furthermore, this study is comparative in nature, and it is important for this aspect to be conveyed in the title. The assessment of MCW and ECI might be regarded as secondary objectives

Several prior research in the literature have examined the assessment of MCI on panoramic radiographs, and this study does not introduce any new findings.

Author Response

Point-to-point responses:

The reviewer’s comments are listed in Italic font. Please find changes to the manuscript highlighted in yellow. A draft of this manuscript was edited Edanz (https://jp.edanz.com/ac): English editing service.

< Detailed Response to Reviewers >

Reviewer: 1

The researchers conducted a study to assess the cognitive impairment in female patients who were taking antiresorptive medicine. The study design is not accurately represented by the title. The authors have assessed MCI, MCW and ECI). The title does not indicate this. Furthermore, this study is comparative in nature, and it is important for this aspect to be conveyed in the title. The assessment of MCW and ECI might be regarded as secondary objectives.

Response: AGREE

Dear Reviewer 1, thank you very much for your detailed review. As pointed by the reviewer, our title was insufficient to reflect the content of the paper. Therefore, we have revised the title as “Radiographical Diagnostic Evaluation of Mandibular Cortical Index Classification and Mandibular Cortical Width in Female Patients Prescribed Antiosteoporosis Medication: A Retrospective Cohort Study”

Changes made: [Page 1, Line 2-5]

Reviewer 2 Report

Comments and Suggestions for Authors

The article titled "Radiographical Evaluation of Mandibular Cortical Index Classification in Female Patients Prescribed Antiosteoporosis Medication" investigates the utility of Digital Panoramic Radiography (DPR) in diagnosing osteoporosis through mandibular cortical index (MCI) classifications, particularly in women undergoing antiosteoporosis medication (AOM). The study is retrospective, utilizing a cohort of 350 female patients aged 40 or older, with a particular focus on comparing those using AOM against a control group not using these medications.

 The topic is highly relevant as it addresses a gap in diagnosing osteoporosis, a prevalent condition with significant health impacts, particularly in asymptomatic patients.

 The study employs a robust methodology, including detailed demographic and medical data analysis, and a comprehensive statistical approach. The use of blinded examiners to assess the MCI reduces bias.

The study provides a thorough statistical examination, including interrater reliability and various tests for evaluating the data, ensuring the robustness of the findings.

The study is limited to a single-center, which may not provide a representative sample of the general population. This limits the generalizability of the findings.

 The significantly younger age of the control group compared to the AOM group could introduce age-related biases in bone density and dental health, which might influence the outcomes.

While the study accounts for several variables, the impact of additional confounders like dietary habits, physical activity levels, and genetic factors on bone health is not discussed.

The study fills a niche in osteoporosis diagnostics, using dental imaging to predict bone health. The methodology is solid, with clear data collection and analysis protocols that align with the study’s objectives. The findings could influence diagnostic practices in both dentistry and general medicine, suggesting a high potential impact on clinical guidelines.

I recommend this article for publication with minor revisions. The authors should consider expanding their discussion on the limitations related to the age differences between the control and AOM groups and possibly the need for a multicenter study to confirm the findings and enhance the generalizability. Further exploration into the long-term effects of AOM on mandibular measurements could also enrich the study’s conclusions.

Overall, the study is well-conducted and presents findings that are valuable to the field of osteoporosis diagnostics, with a novel approach linking dental health to systemic bone conditions.

Author Response

Point-to-point responses:

The reviewer’s comments are listed in Italic font. Please find changes to the manuscript highlighted in yellow. A draft of this manuscript was edited Edanz (https://jp.edanz.com/ac): English editing service.

< Detailed Response to Reviewers >

Reviewer: 2

Thank you very much for your valuable and constructive comments. Your remarks are well-founded and gave us a better understanding how our manuscript can be improved.

The article titled "Radiographical Evaluation of Mandibular Cortical Index Classification in Female Patients Prescribed Antiosteoporosis Medication" investigates the utility of Digital Panoramic Radiography (DPR) in diagnosing osteoporosis through mandibular cortical index (MCI) classifications, particularly in women undergoing antiosteoporosis medication (AOM). The study is retrospective, utilizing a cohort of 350 female patients aged 40 or older, with a particular focus on comparing those using AOM against a control group not using these medications. The topic is highly relevant as it addresses a gap in diagnosing osteoporosis, a prevalent condition with significant health impacts, particularly in asymptomatic patients. The study employs a robust methodology, including detailed demographic and medical data analysis, and a comprehensive statistical approach. The use of blinded examiners to assess the MCI reduces bias. The study provides a thorough statistical examination, including interrater reliability and various tests for evaluating the data, ensuring the robustness of the findings. The study is limited to a single-center, which may not provide a representative sample of the general population. This limits the generalizability of the findings.

Response: AGREE

Dear Reviewer 2, thank you very much for your kindly comment. We are glad that you understand the purpose of our appeal. As the reviewer noted, the sample we used was limited and there were difficulties in generalizing our conclusions. This point has been added to the limitation. In the future, we would like to design a large-scale survey to address this issue.

Changes made: [Page 10, Line 365-367]

 The significantly younger age of the control group compared to the AOM group could introduce age-related biases in bone density and dental health, which might influence the outcomes. While the study accounts for several variables, the impact of additional confounders like dietary habits, physical activity levels, and genetic factors on bone health is not discussed.

Response: AGREE

Dear Reviewer 2, thank you very much for your constructive comment. As the reviewer mentioned, there was a large gap between the age of the test group and the control. However, this cohort was selected for patients attending the same facility for the purpose of eliminating bias among patients. Therefore, we believe that we were able to reduce the bias of the surgeons and testing methods. Eliminating all biases is a difficult problem in any study. The results obtained were taken seriously and this point was added to the discussion. In addition, although the reviewers recommended various important confounding factors, none of them could be investigated in all patients, and complete data were not available. Of course, we also believe that incorporating these variables will allow more accurate study results to be presented. We appreciate your advice.

Changes made: [Page 8, Line 272-278]

The study fills a niche in osteoporosis diagnostics, using dental imaging to predict bone health. The methodology is solid, with clear data collection and analysis protocols that align with the study’s objectives. The findings could influence diagnostic practices in both dentistry and general medicine, suggesting a high potential impact on clinical guidelines. I recommend this article for publication with minor revisions. The authors should consider expanding their discussion on the limitations related to the age differences between the control and AOM groups and possibly the need for a multi-center study to confirm the findings and enhance the generalizability. Further exploration into the long-term effects of AOM on mandibular measurements could also enrich the study’s conclusions. Overall, the study is well-conducted and presents findings that are valuable to the field of osteoporosis diagnostics, with a novel approach linking dental health to systemic bone conditions.

Response: AGREE

Dear Reviewer 2, thank you very much for your detailed review. It is very meaningful that you agree with the results of the study we have described. We have responded to the advice you have given us as described above. Our paper is now logically robust and we thank you.

Reviewer 3 Report

Comments and Suggestions for Authors

Radiographical Evaluation of Mandibular Cortical Index Classification in Female Patients Prescribed Antiosteoporosis Medi cation

This paper is addressed to the radiographic findings in patients using antiosteoporosis medication (AOM) and aimed to investigate the influence of the mandibular cortical index (MCI) classification on the osteoporosis diagnosis results. The study is not adequately designed and there are methodological failures. In addition, there are studies already addressed to this topic,  and I am afraid that this study does not bring new information.

Abstract

The background does not support the emerging aim.

The Aim should be rewritten. According to the Conclusion, I suppose that this study aimed to assess the sensitivity and specificity of MCI in the diagnosis of osteoporosis in following patient groups: using AOM and not using AOM.

The Results are not clearly and completely written.

Introduction  

The clinical relevance of the study is not well clarified. The authors aimed to detect characteristic radiological findings in patients using AOM to enable better diagnosis. What diagnosis? This patients are already diagnosed and on AOM treatment.

Material and methods

Women aged 40 years and older who had digital panoramic radiographs met the inclusion criteria. The average age for menopause is 50 years. It can start in 40s, but manifested osteoporosis is more likely to be diagnosed over 50 years. Adjusting the inclusion criteria to align with the typical onset of osteoporosis could reduce the age differences between the examined and the control group.

To answer the set goal, you should form three different groups: 1) women with diagnosed osteoporosis on AOM, 2) women with diagnosed osteoporosis without AOM, and control group consisting of healthy women.  The sample size should be determined using the power analysis, and calculated for each group to be almost equaled, with defined: effect size, the margin of error, confidence level, and universe representation power.  The control group in the presented study consisted of 301 participants, of which only 5 had diagnosed osteoporosis. Thus, you cannot draw a conclusion as you already did, if you rely on this sample.

This study included patients that used bisphosponates for reasons other than osteoporosis.  These patients should be excluded.

Additionally, patients with metabolic bone disease other than osteoporosis (hyper- or hypoparathyroidism, Paget disease, osteomalacia, renal osteodystrophy, or osteogenesis imperfecta) should also be excluded from the study.

In summary, refining the study’s objectives, clarifying clinical relevance, and addressing methodological gaps will enhance the paper’s impact.

Comments on the Quality of English Language

Extensive English editing is needed.

Author Response

Point-to-point responses:

The reviewer’s comments are listed in Italic font. Please find changes to the manuscript highlighted in yellow. A draft of this manuscript was edited Edanz (https://jp.edanz.com/ac): English editing service.

< Detailed Response to Reviewers >

Reviewer: 3

Radiographical Evaluation of Mandibular Cortical Index Classification in Female Patients Prescribed Antiosteoporosis Medication

This paper is addressed to the radiographic findings in patients using antiosteoporosis medication (AOM) and aimed to investigate the influence of the mandibular cortical index (MCI) classification on the osteoporosis diagnosis results. The study is not adequately designed and there are methodological failures. In addition, there are studies already addressed to this topic, and I am afraid that this study does not bring new information.

Response: AGREE

Dear Reviewer 3, thank you very much for your valuable and constructive comments. Your remarks are well-founded and gave us a better understanding how our manuscript can be improved. Corrections were made regarding deficiencies in the study design, which were also mentioned by other reviewers. Please see the comments below for the corrections. On the other hand, with regard to the novelty of the paper, other reviewers evaluated this study as presenting new and valuable findings for the field of osteoporosis diagnosis. As Reviewer 3 point out, this diagnostic method has been used for a long time, but we believe that there are still few studies on the findings in AOM users. The following statement was added as a strength of the paper in this regard:

Although screening diagnostic methods using MCI were discovered about 20 years ago, few studies have examined imaging findings in AOM users, which is a novel point of view.

Changes made: [Page 10, Line 360-362]

Abstract

The background does not support the emerging aim. The Aim should be rewritten. According to the Conclusion, I suppose that this study aimed to assess the sensitivity and specificity of MCI in the diagnosis of osteoporosis in following patient groups: using AOM and not using AOM. The Results are not clearly and completely written.

Response: AGREE

We thank the reviewer for pointing this out. As noted by the reviewer, the purpose of the research objectives and conclusions was poor. The following new text has been changed:

・The purpose of this study was to evaluate the sensitivity and specificity of mandibular cortical index (MCI) in the diagnosis of osteoporosis in a group of patients with and without antiosteoporosis medication (AOM).

・This method of estimating osteoporosis based on MCI classification using DPR images has high sensitivity, especially in patients using AOM, suggesting that this method is useful as a screening test.

Changes made: [Page 1, Line 23-26, 32-34]

Introduction

The clinical relevance of the study is not well clarified. The authors aimed to detect characteristic radiological findings in patients using AOM to enable better diagnosis. What diagnosis? This patients are already diagnosed and on AOM treatment.

Response: AGREE

We thank the reviewer for the deep insight. Reviewer 3's comments are logical. Our claim to "diagnosis" has two meanings. The first is to diagnose osteoporosis itself by the ravaging findings of the DPR images. But we know that this is a totally challenging idea and only hints at a formal diagnosis. The second is the name of the disease of osteoporosis as officially diagnosed by the physician. This study adopted this result as the gold standard. For this reason, of course, many cases are treated with AOM. However, there were also many untreated patients who were overlooked. The motivation for this study is for dentists to find these undiagnosed osteoporosis patients, which is described in the introduction.

Material and methods

Women aged 40 years and older who had digital panoramic radiographs met the inclusion criteria. The average age for menopause is 50 years. It can start in 40s, but manifested osteoporosis is more likely to be diagnosed over 50 years. Adjusting the inclusion criteria to align with the typical onset of osteoporosis could reduce the age differences between the examined and the control group.

To answer the set goal, you should form three different groups: 1) women with diagnosed osteoporosis on AOM, 2) women with diagnosed osteoporosis without AOM, and control group consisting of healthy women. The sample size should be determined using the power analysis, and calculated for each group to be almost equaled, with defined: effect size, the margin of error, confidence level, and universe representation power.  The control group in the presented study consisted of 301 participants, of which only 5 had diagnosed osteoporosis. Thus, you cannot draw a conclusion as you already did, if you rely on this sample.

This study included patients that used bisphosponates for reasons other than osteoporosis. These patients should be excluded. Additionally, patients with metabolic bone disease other than osteoporosis (hyper- or hypoparathyroidism, Paget disease, osteomalacia, renal osteodystrophy, or osteogenesis imperfecta) should also be excluded from the study.

Response: AGREE

Dear Reviewer 3, thank you very much for your detailed review. Thank you for pointing out the study design and for your advice on patient selection criteria. Other reviewers also pointed out the age gap between the AOM and control groups. This was analyzed as a detrimental effect of the reason that the experiment was conducted at a single facility in order to reduce bias among surgeons and medical facilities. We also set the lower age limit for disease coverage at 40 years old instead of 50 years old because we thought we could expect a larger sample. The number of osteoporosis patients shown in Table 1 is the number diagnosed by the physician (AOM group: 41; control group: 5). The sample size calculation was performed with a sample ratio of 0.84 for the AOM group, 0.01 for the control group, an alpha error of

0.05, power 0.8, and sample ratio of the two groups was 1:0.12, these were added to the Materials and Methods. It is our regret that we were not able to extract all the reasons for the use of AOM (those that were not osteoporosis) from the sample. This was uncertain information due to the lack of a thorough medical interview. To see the effect of pure osteoporosis, it is necessary to exclude other metabolic diseases, as the reviewer points out. Since this is an essential consideration for future research, it was added to the limitation.

Changes made: [Page 5, Line 168-170]

・The sample size calculation was based on the number of people diagnosed with osteoporosis at hospitals, with an alpha error of 0.05, a power of 0.8, and a sample ratio of 1:0.12 for the two groups.

Changes made: [Page 10, Line 368-370]

・Additionally, in future studies, to examine the effect of pure osteoporosis only, it is necessary to exclude AOM patients used for non-osteoporotic purposes and patients with other metabolic diseases.

In summary, refining the study’s objectives, clarifying clinical relevance, and addressing methodological gaps will enhance the paper’s impact.

Response: AGREE

Finally, we would like to thank you for your valuable efforts to improve the quality of this manuscript. We believe that your comment significantly improved the manuscript and will help readers interpret and implement the knowledge transferred in the present manuscript.

Round 2

Reviewer 3 Report

Comments and Suggestions for Authors

Dear authors,

Expecting a larger sample is not a justification for adjusting the age limit for disease. You should take a longer time span.

Lines 373-374: “This method of estimating osteoporosis based on MCI classification using DPR images has high sensitivity, especially in patients using AOM, suggesting that this method is useful as a screening test.“ Erase especially because you had only 5 patients with osteoporosis and not using AOM.

You should point out the importance of determination of the “cut off value” when using MCI in the Conclusion section, having in mind its impact on the sensitivity. That will also contribute to the novelty.

Comments on the Quality of English Language

No comment

Author Response

Point-to-point responses:

The reviewer’s comments are listed in Italic font. Please find changes to the manuscript highlighted in yellow.

< Detailed Response to Reviewers >

Reviewer: 3

Expecting a larger sample is not a justification for adjusting the age limit for disease. You should take a longer time span.

Response: AGREE

First, we thank reviewer 3 for kindly reviewing our paper several times. Reviewer 3 provided constructive comments on the importance of increasing the sample size and methodology. As suggested by reviewer 3, we would like to increase the duration of the experiment and improve recruitment of subjects in future studies. The following statement was further added to the limitation:

Future studies should improve this problem with longer experimental duration.

Changes made: [Page 10, Line 361]

Lines 373-374: “This method of estimating osteoporosis based on MCI classification using DPR images has high sensitivity, especially in patients using AOM, suggesting that this method is useful as a screening test.“ Erase especially because you had only 5 patients with osteoporosis and not using AOM

Response: AGREE

We thank the reviewer for pointing this out. The reviewer's points are logical. We agree with the reviewer and have removed some of the conclusions.

Changes made: [Page 10, That sentence in the conclusion was deleted.]

You should point out the importance of determination of the “cut off value” when using MCI in the Conclusion section, having in mind its impact on the sensitivity. That will also contribute to the novelty

Response: AGREE

Dear Reviewer 3, we appreciate your very kind comments. We received deep insight into the points that should be addressed in the conclusion. Based on this advice, the sentences were modified as follows:

This diagnostic method requires careful determination of cut off values using the severity of the classifications. New findings show that changes in the cutoff value affect the sensitivity of the test.

Changes made: [Page 10, Line 368-370]

Finally, we would like to thank you for your valuable efforts to improve the quality of this manuscript. We believe that your comment significantly improved the manuscript and will help readers interpret and implement the knowledge transferred in the present manuscript.

Round 3

Reviewer 3 Report

Comments and Suggestions for Authors

I appreciate the changes  you made.